# Electropositive Membrane Prepared via a Simple Dipping Process: Exploiting Electrostatic Attraction Using Electrospun SiO_2_/PVDF Membranes with Electronegative SiO_2_ Shell

**DOI:** 10.3390/polym15102270

**Published:** 2023-05-11

**Authors:** Dalsu Choi, Cheol Ho Lee, Han Bi Lee, Min Wook Lee, Seong Mu Jo

**Affiliations:** 1Chemical Engineering Department, Myongji University, Yongin-si 17058, Gyeonggi-do, Republic of Korea; dalsuchoi@mju.ac.kr; 2Center for Underground Physics, Institute for Basic Science, Daejeon 34126, Republic of Korea; lch2301@ibs.re.kr; 3Composite Materials Applications Research Center, Korea Institute of Science and Technology, Wanju-gun 55324, Jeollabuk-do, Republic of Korea; 092414@kist.re.kr

**Keywords:** electrospinning, electrostatic attraction, dipping, electropositive membrane, filter

## Abstract

This research aimed to develop a simple and cost-effective method for fabricating electropositive membranes for highly efficient water filtration. Electropositive membranes are novel functional membranes with electropositive properties and can filter electronegative viruses and bacteria using electrostatic attraction. Because electropositive membranes do not rely on physical filtration, they exhibit high flux characteristics compared with conventional membranes. This study presents a simple dipping process for fabricating boehmite/SiO_2_/PVDF electropositive membranes by modifying an electrospun SiO_2_/PVDF host membrane using electropositive boehmite nanoparticles (NPs). The surface modification enhanced the filtration performance of the membrane, as revealed by electronegatively charged polystyrene (PS) NPs as a bacteria model. The boehmite/SiO_2_/PVDF electropositive membrane, with an average pore size of 0.30 μm, could successfully filter out 0.20 μm PS particles. The rejection rate was comparable to that of Millipore GSWP, a commercial filter with a pore size of 0.22 μm, which can filter out 0.20 μm particles via physical sieving. In addition, the water flux of the boehmite/SiO_2_/PVDF electropositive membrane was twice that of Millipore GSWP, demonstrating the potential of the electropositive membrane in water purification and disinfection.

## 1. Introduction

Currently, owing to its energy efficiency, pressure-driven membrane filtration is the most widely used technique for water purification [1,2]. Among various membrane-filtration approaches, physical sieving is the most common technique [1,3]. Although they are easily fabricated, filters based on physical sieving have a “flux–pore-size” issue when used to filter ultrafine particles, which require ultrafine pore sizes [4]. The reduction of flux is inevitably accompanied by pore size reduction. Furthermore, considerable pressure must drive ultrafine pore membranes, which reduces filtration efficiency [5,6]. Currently, various techniques, such as mechanical and chemical methods, are being investigated for filtration. Water purification using bulk mechanical filters is the most common technique in water treatment. Sand, hydro-anthracite, burned rocks, and crushed expanded clay are also used for filtration [7]. Additionally, various types of adsorption membranes have been used to remove heavy metals and organic dyes from wastewater [8]. Among them, metal–organic frameworks (MOFs) have been used in many fields owing to their high surface area and tunable pore volume and chemical properties, and MOF mixed membranes containing nanoparticles (NPs) have been studied [9]. Recently, membrane filtration based on mechanisms other than physical sieving has attracted considerable attention [5,10,11,12,13,14].

Electropositive filtration is a technique that uses a filtration mechanism other than sieving. An electropositive filter is fabricated by depositing highly electropositive components on the outer surface of the host membrane [15]. Polyelectrolytes [16], zirconia [17], copper oxide [18], and hematite [19] have been recently investigated as electropositive coatings. However, such electropositive coatings usually suffer from poor adhesion to the membrane, low zeta potential, and toxicity. Aluminum oxide, gibbsite, and activated alumina are the most widely used electropositive components for electropositive membranes owing to their low toxicity, low cost, and high mechanical and thermal stability [11,20,21,22]. By incorporating electropositive components, electronegative substances in a feed, such as bacteria and viruses, which commonly show a negative charge in water media, can be effectively filtered via electrostatic attraction [20,23,24]. Therefore, ultrafine electronegative particles can be filtered through pores larger than the particles without sacrificing the flux of the feed. However, the attachment of boehmite on membrane surfaces requires complicated processes, such as the direct hydrothermal synthesis of boehmite on the host membrane [14,21,25]. To date, boehmite deposition during the fabrication of electropositive membranes is achieved by direct hydrothermal synthesis on the host membrane [9]. Therefore, polymeric hosts cannot be used, as they cannot withstand harsh hydrothermal conditions. A simple dipping process for boehmite deposition would enable the usage of various host membranes. In addition, dipping processes allow continuous fabrication, making them more cost-effective than batch-type hydrothermal processes. PVDF-g-PNE and PVDF-g-PAA membranes contain electropositive materials; thus, bacteria and viruses, which normally exhibit negative charges, can be effectively filtered via electrostatic attraction. Polymer coatings firmly bond to PVDF membranes through adhesive force (coordination, hydrogen bonding, electrostatic interaction, and hydrophobic interaction) [26]. SiO_2_ NPs have high electronegativity (2.82) and attract electropositive lithium ions from electrolytes. On this basis, a SiO_2_/PVDF composite membrane was developed as a battery separator by varying the SiO_2_-to-PVDF mass ratio [27]. Graft copolymerization of methacrylic acid (MAA) monomers with plasma-activated PVDF membranes was performed to introduce carboxyl groups into the membrane. Subsequently, the surface of the NPs was made hydrophilic using a positively charged ligand, and the NPs were coated on the membrane surface through electrostatic and covalent bonding [28].

Here, a simple technique for attaching boehmite to host membranes is proposed. The performance of the resultant boehmite/SiO_2_/PVDF electropositive membrane was evaluated and compared with that of commercially available filters. The obtained boehmite/SiO_2_/PVDF electropositive membrane exhibited better performance than the commercial filters. The water flux of the boehmite/SiO_2_/PVDF electropositive membrane was almost twice that of Millipore GSWP, a conventional filter with a pore size similar to that of the fabricated membrane, and the particle rejection rates of both membranes were comparable.

## 2. Materials and Methods

### 2.1. Materials

Tetraethyl orthosilicate (TEOS), aluminum isopropoxide (AIP), and acetic acid were purchased from Sigma-Aldrich (St. Louis, MO, USA). N,N-dimethylformamide (DMF, 99.8%), ethyl alcohol (EtOH, 94.5%), hydrochloric acid (HCl, 37%), and sodium hydroxide (NaOH) were purchased from Daejung Reagent Chemicals, and poly(vinylidene fluoride) (PVDF, Kynar-761) was purchased from Alkema. All chemicals were used as received without further purification. Uniform polystyrene (PS) latex beads (0.20 μm) used for particle rejection tests were obtained from Magsphere Inc. Millipore GSWP with a pore size of 0.22 μm was used as a commercial filter to compare with the boehmite/SiO_2_/PVDF electropositive membrane.

### 2.2. Synthesis of Boehmite (γ–AlOOH)

Boehmite (γ–AlOOH) was synthesized via the sol–gel reaction of aluminum isopropoxide (AIP). First, 68 g of AIP was added to the 300 mL of deionized water at 75 °C. The solution was heated to 95 °C with mechanical stirring, and the water was evaporated until the volume of the solution was reduced to 200 mL. Next, 3.1 g of acetic acid was added dropwise, and the solution was stirred for 10 min. Finally, hydrothermal synthesis was performed at 150 °C for 6 h using an autoclave.

### 2.3. Preparation of SiO_2_/PVDF Solution

The silica precursor was prepared by the sol–gel reaction of TEOS. H_2_O and HCl were added to the TEOS/EtOH solution in a TEOS-EtOH-H_2_O-HCl molar ratio of 0.54:1.08:1.08:0.005. The prepared mixture was heated at 95 °C for 2 h to accelerate the condensation polymerization, after which it was cooled to room temperature, and 100 g of DMF was added before irreversible gelation occurred. The prepared silica sol was mixed with PVDF in a silica sol–PVDF weight ratio of 3:7. Then, DMF was added to the prepared silica sol/PVDF/DMF solution with 19.44 weight percent of silica sol/PVDF. Finally, by stirring the silica sol/PVDF/DMF solution at 100 °C for 1 h, a SiO_2_/PVDF blend solution was obtained.

### 2.4. Preparation of Electrospun SiO_2_/PVDF Membrane

SiO_2_/PVDF membranes were fabricated by electrospinning the SiO_2_/PVDF solution. The SiO_2_/PVDF solution was fed at a rate of 5 μL/min through a 24 G needle, and the distance from the needle tip to the substrate was set to 15 cm. Then, 13 kV voltage was applied at a relative humidity of 18 ± 2% and a temperature of 30 ± 1 °C. Through a 4 h collection of the electrospun fibers, a 30-μm-thick membrane was obtained.

### 2.5. Preparation Boehmite/SiO_2_/PVDF Membrane

Boehmite was attached to an electrospun SiO_2_/PVDF membrane by submerging the membrane in an 11.5 wt% boehmite solution for 10 min. After the dipping process, the boehmite/SiO_2_/PVDF composite membrane was washed by flowing 500 mL of deionized water through the membrane using a filter funnel. Finally, the washed membrane was dried in a convection oven at 60 °C for 6 h. The amount of attached boehmite was determined by comparing the weight of the boehmite-attached SiO_2_/PVDF membrane with that of the bare SiO_2_/PVDF membrane using a microbalance.

### 2.6. Membrane Performance Tests

The pore size distribution and average pore size of the electrospun SiO_2_/PVDF and boehmite/SiO_2_/PVDF membranes were measured using a capillary flow porometer (CFP-1500AE, Porous Materials Inc., Ithaca, NY, USA) following the ASTM-F316-03 standard. The membranes were soaked with a Galwick™ wetting liquid with a surface tension of 15.9 dyn/cm, and air was passed through the membrane. For the water flux test, deionized water was fed through a dead-end cell (AMICON stirred cell 8010) by applying pressure to the reservoir using nitrogen gas. The pure water flux was calculated by measuring the weight of deionized water passed through the membrane inside the dead-end cell using a microbalance (XS6002S, Mettler-Toledo, Columbus, OH, USA). Next, a particle rejection test was performed using a setup similar to that of the water flux test. First, the membrane for the rejection test was assembled inside the dead-end cell (AMICON stirred cell 8010). Then, 100 ppm of 0.20 μm PS particles dispersed in deionized water was fed into the dead-end cell at 2.5 psi for 30 min. Then, the permeate was collected at intervals, and the collected permeates were analyzed by ultraviolet–visible (UV–Vis) spectroscopy (V-670, Jasco, Portland, OR, USA). The characteristic absorbance peak of PS NPs at 230 nm was monitored to determine the concentration of the NPs.

### 2.7. Characterization

The crystal structure of the synthesized boehmite and electrospun SiO_2_/PVDF membrane was analyzed using an X-ray diffractometer (Rigaku, Tokyo, Japan) with a Cu Kα (λ = 0.154 nm) beam source. Samples were scanned from 2θ = 10° to 90°, and the instrument was driven at the acceleration voltage of 45 kV and emission current of 200 mA. The morphology of the synthesized boehmite and the cross-section of the electrospun SiO_2_/PVDF membrane were analyzed by field-emission transmission electron microscopy (FE-TEM, Tecnai F20, FEI, Hillsboro, OR, USA). A boehmite solution was drop cast on a copper TEM grid, and the grid was thoroughly dried in a vacuum oven before the TEM measurement. To measure the cross-section of the electrospun SiO_2_/PVDF membrane, the fabricated membrane was cross-sectioned using a Cryo-Ultramicrotome (RMC-PTPC + CR-X), and the prepared sample was put on a copper TEM grid for measurements. Dynamic laser scattering (DLS, Malvern Zetasizer, Malvern, UK) analysis was used to determine the zeta potential of the samples at different pH values and at the isoelectric point (IEP). The synthesized boehmite and 0.20 µm PS particles were dispersed in ethyl alcohol to make a 100 ppm dispersion for the DLS measurements. Then, 0.1 mol/L NaOH and 0.1 mol/L HCl solutions were added to vary the pH of the dispersion, and the pH was monitored in situ using a pH meter (Orion Star A211, Thermo Scientific, Waltham, MA, USA). The morphologies of the electrospun SiO_2_/PVDF and boehmite/SiO_2_/PVDF composite membranes were analyzed by FE-scanning electron microscopy (SEM) (Nova NanoSEM450, FEI, Hillsboro, OR, USA) at an acceleration voltage of 10 kV. The atomic composition and presence of specific chemical bonds were studied using X-ray photoelectron spectroscopy (XPS) with an Al K-alpha X-ray source (K-Alpha X-ray Photoelectron Spectrometer System, Thermo Fisher Scientific, Waltham, MA, USA). For the survey scan, 200 eV pass energy was used, and the scanning was performed twice. For fine scans, 50 eV pass energy was used, and five scans were performed. All scans were executed with a flood gun to minimize the charge accumulation. The resultant spectra were analyzed using the Advantage software provided with the XPS instrument. To check the amount of boehmite detached from the boehmite/SiO_2_/PVDF composite membrane during filtration, the filtrate was analyzed using inductively coupled plasma–optical emission spectroscopy (ICP–OES, iCAP 6000, Thermo Scientific, Loughborough, UK). Furthermore, 100, 300, and 500 mL of deionized water were filtered through the boehmite/SiO_2_/PVDF composite membrane, and the aluminum contents of filtrates were measured. The contact angles of the membranes were determined using a contact angle and surface tension analyzer (Phoenix 300).

## 3. Results

### 3.1. Characterization of the Synthesized Boehmite (γ–AlOOH)

Figure 1a shows the TEM image of the synthesized boehmite. It reveals an assembly of nanorods, which is a characteristic feature of boehmite [14,21,22]. Furthermore, the X-ray diffraction (XRD) pattern of the sample (Figure 1b) showed diffraction peaks at 2θ = 14.50°, 28.24°, 38.44°, and 49.18°, which are ascribed to the (020), (120), (031), and (200) planes of boehmite, respectively [22], confirming the successful synthesis of boehmite nanorods. Finally, the synthesized boehmite exhibited a highly positive zeta potential ranging from 30 to 50 mV under acidic conditions (Figure 1c). Even at a neutral pH, the zeta potential was 32.8 mV. A highly positive zeta potential was maintained from low pH values to the IEP (pH 9), where the zeta potential became zero. Therefore, the sample would exhibit an electropositive effect for filtration in a wide range of environments.

### 3.2. Characterization of Electrospun SiO_2_/PVDF Membrane

The morphology and nanofiber dimensions of the electrospun SiO_2_/PVDF membrane were characterized by SEM. The average diameter of the SiO_2_/PVDF nanofibers in the membrane was 134 ± 43 nm. Furthermore, a well-developed internetworking structure with mesopores was observed, indicating the formation of electrospun nanofibers.

Next, the atomic composition and available chemical bonds of the electrospun SiO_2_/PVDF membrane surface were determined by XPS to investigate the presence of SiO_2_. Si 2p and O 1s scans (Figure 2b,c) showed peaks at the binding energies of 103.7 and 533.2 eV, which were ascribed to the Si–O bond of SiO_2_ [29], indicating SiO_2_. In addition, the atomic ratio of Si:O decreased from 24.4:44.4 to 1:2 (Table 1), further indicating SiO_2_ in the electrospun SiO_2_/PVDF nanofiber. The XPS survey spectra (Figure 2a) showed Si 2p, C 1s, O 1s, and F 1s, which are atomic components of SiO_2_ and PVDF. This shows that at an XPS penetration depth of up to 10–20 nm, PVDF and SiO_2_ were present in the sample. However, as the combined atomic percentage of carbon and fluoride from PVDF is much lower than that of silicon and oxide from SiO_2_, the outer portion of the electrospun SiO_2_/PVDF fiber mainly consists of SiO_2_ [30]. Thus, the electrospun SiO_2_/PVDF fiber was cross-sectioned using ultramicrotomy, and the prepared sample was analyzed by TEM (Figure 3). Consistent with the XPS result, the outermost part of the electrospun SiO_2_/PVDF fiber comprised SiO_2_. Additionally, the TEM image revealed the formation of multicore structures, which was attributed to the distinct phase separation induced by unfavorable molecular interactions between the SiO_2_ sol and PVDF.

### 3.3. Characterization of Boehmite/SiO_2_/PVDF Composite Membrane

Electrospun SiO_2_/PVDF membranes with highly electronegative SiO_2_ skin can be effective in preparing electropositive membranes because highly electropositive boehmite can be easily attached to a membrane via electrostatic interactions [31]. Here, the electrospun SiO_2_/PVDF membrane was simply submerged in a solution of the synthesized boehmite for 10 min to enable the attachment of the boehmite to the membrane via electrostatic attractions. Via a simple dipping process, boehmite was successfully attached to and fully covered the SiO_2_/PVDF membrane, whereas in previous studies, intense hydrothermal treatment has been required to attach boehmite directly to the surfaces of host membranes. The SEM images of the SiO_2_/PVDF membrane after the simple, short dipping process (Figure 4a,b) revealed the successful attachment of boehmite to the membrane. The SiO_2_/PVDF fibers showed a smooth surface (panel a), whereas an irregular texture was observed on the fiber surface (panel b), confirming that boehmite coated the SiO_2_/PVDF fibers. The increased membrane weight after the dipping process also indicated the successful attachment of boehmite to the membrane. On average, 12.65 wt% of boehmite was attached after dipping (Appendix A). Boehmite is mainly composed of aluminum. Although instant exposure to aluminum is not critical to human beings, prolonged exposure to aluminum can cause health issues [32]. Thus, the amount of detached boehmite after filtration was examined to ensure that no aluminum was introduced into the filtrate. First, 100, 300, and 500 mL deionized water samples were filtered through the electrospun boehmite/SiO_2_/PVDF membrane, and the filtrates were analyzed by ICP–OES. The amount of aluminum in the three samples was undetectable, as it was less than 0.1 ppm (Figure 4c).

### 3.4. Membrane-Filtration Performance

Pore size and pore size distribution, which are critical parameters that determine membrane retention capability, were evaluated using a capillary flow porometer (Figure 5a). The mean pore diameter of the electrospun SiO_2_/PVDF membrane decreased from 0.48 to 0.35 μm upon the attachment of boehmite, which was attributed to the increased diameter of the nanofiber upon boehmite attachment. In addition, the mean pore size of Millipore GSWP, a commercial MF membrane with a smaller pore size, is 0.22 μm, which is smaller than that of the boehmite/SiO_2_/PVDF composite. To determine the water flux efficiency of the boehmite/SiO_2_/PVDF composite membrane, a dead-end cell system was assembled to measure the water flux. Millipore GSWP showed no considerable change in flux upon operation, whereas the flux of the electrospun SiO_2_/PVDF and boehmite/SiO_2_/PVDF composite membranes decreased upon operation, which was attributed to the water-flow-driven compression of the less densely packed electrospun membranes. The boehmite/SiO_2_/PVDF membrane showed considerably improved water flux compared with the SiO_2_/PVDF membrane (Figure 5b,c). The water flux through the SiO_2_/PVDF membrane was 6463 L∙m^−2^∙h^−1^_,_ whereas that of the boehmite/SiO_2_/PVDF composite membrane was 18,827 L∙m^−2^∙h^−1^. This notable enhancement in water flux in the composite membrane can be attributed to the hydrophilicity of the membrane, which provides an additional driving force for water to easily pass through the capillary induced by membrane pores [29,30,31,33,34,35,36,37,38,39,40,41]. To verify our hypothesis, we measured the water contact angles of the boehmite/SiO_2_/PVDF composite and SiO_2_/PVDF membranes (Figure 5d). Due to the aforementioned hydrophilicity, the membrane with boehmite had a much lower contact angle than the SiO_2_/PVDF membrane (37.92° vs. 127.85°). In the case of low-pressure conditions (2.5 psi), the water flux of the electrospun membranes was higher than that of Millipore GSWP.

Finally, to evaluate the rejection ability of the boehmite/SiO_2_/PVDF composite membrane, particle rejection tests were conducted using 0.20 ± 0.011 μm PS particles. The filtration efficiency of each membrane was measured for 30 min under a pressure of 2.5 psi by filtering a 100 ppm PS dispersion sample. The size of common bacteria was modeled by 0.20 μm particles. Additionally, the zeta potential of the PS particles was measured by DLS analysis (Figure 6a). The PS particles showed a highly negative zeta potential (−46.4 mV) at pH 7. Therefore, we concluded that the filtration of 0.20 μm PS NPs can effectively simulate bacteria rejection. The particle rejection percentage for the tested membranes increased with the filtration time due to the pore-blocking effect of the NPs. However, the boehmite/SiO_2_/PVDF membrane with a large pore size and a considerably high water flux exhibited a high rejection rate similar to that of Millipore GSWP (Figure 6b). Considering the high water flux, which was twice that of Millipore GSWP, the boehmite/SiO_2_/PVDF membrane exhibited better performance than Millipore GSWP. Therefore, the adoption of boehmite aids in more effective filtration via electrostatic attraction and enhancing interface interactions.

## 4. Conclusions

This study presents a simple, fast, cost-effective technique for fabricating electropositive membranes. Unlike conventional techniques that require intense hydrothermal reactions to attach electropositive boehmite to a membrane matrix, using the proposed technique, a dense layer of electropositive boehmite can be easily assembled on the surfaces of the electrospun membrane by simply dipping the membrane in a boehmite solution. Furthermore, the performance of the fabricated boehmite/SiO_2_/PVDF membrane was evaluated and compared with that of electrospun membranes and Millipore GSWP filter. The water flux of the boehmite/SiO_2_/PVDF membrane was twice that of the Millipore GSWP filter, which has a smaller pore size than the boehmite/SiO_2_/PVDF membrane. The higher water flux of the boehmite/SiO_2_/PVDF membrane was attributed to the improved interface interactions between the membrane and water upon the attachment of the hydrophilic boehmite. The boehmite/SiO_2_/PVDF membrane showed a very high rejection rate comparable to that of Millipore GSWP. In existing surface treatment methods, pores on the treated surfaces are blocked, thereby lowering the filtration efficiency. The proposed technique overcomes this problem. The low water contact angle of the boehmite/SiO_2_/PVDF increased the flow rate and reduced energy consumption. The boehmite/SiO_2_/PVDF electropositive membrane also outperformed the commercial Millipore GSWP filters owing to the hydrophilicity of boehmite; thus, it is promising for applications in water purification and disinfection.

## Figures and Tables

**Figure 1 polymers-15-02270-f001:**
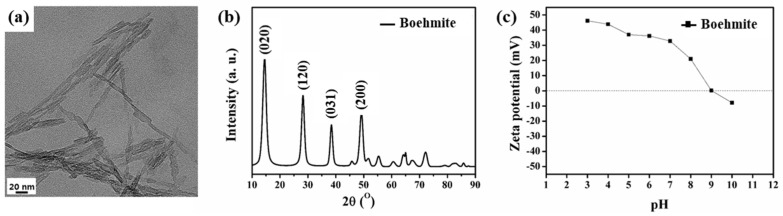
(**a**) Transmission electron microscopy (TEM) image of boehmite synthesized via hydrothermal treatment. (**b**) X-ray diffraction (XRD) pattern of as-prepared boehmite. (**c**) Zeta potential of the synthesized boehmite at pH 3–10.

**Figure 2 polymers-15-02270-f002:**
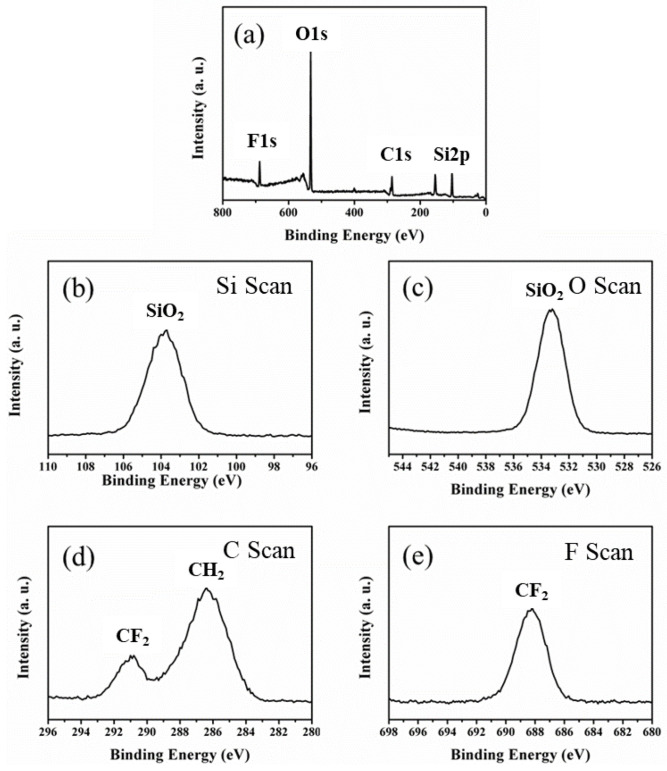
X-ray photoelectron spectroscopy (XPS) images of the electrospun SiO_2_/PVDF membrane. (**a**) Wide scan spectra, (**b**) Si 2p spectra, (**c**) O 1s spectra, (**d**) C 1s spectra, and (**e**) F 1s spectra of the electrospun SiO_2_/PVDF membrane.

**Figure 3 polymers-15-02270-f003:**
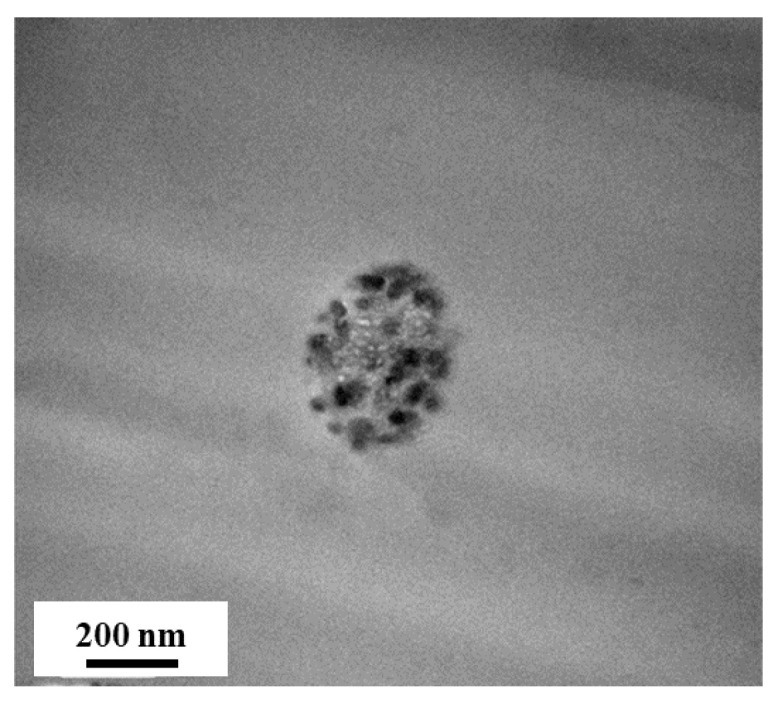
Cross-sectional TEM image of the multicore structure of the electrospun SiO_2_/PVDF blend nanofiber.

**Figure 4 polymers-15-02270-f004:**
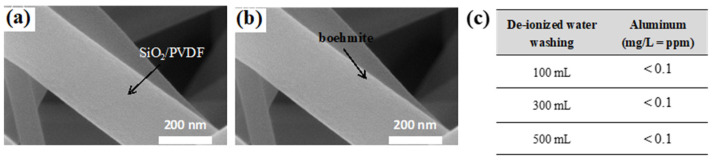
Scanning electron microscopy (SEM) images of (**a**) electrospun SiO_2_/PVDF membrane and (**b**) boehmite-SiO_2_/PVDF composite membrane fabricated by dipping. (**c**) Inductively coupled plasma–optical emission spectroscopy (ICP–OES) results of filtrates of deionized water samples.

**Figure 5 polymers-15-02270-f005:**
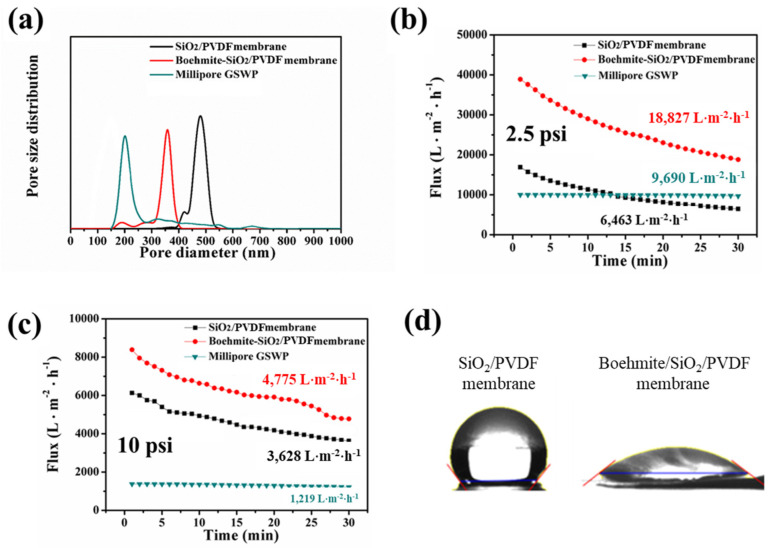
(**a**) Pore size distribution of the fabricated membranes (SiO_2_/PVDF and boehmite/SiO_2_/PVDF) and Millipore GSWP. (**b**,**c**) Water flux of each membrane at the pressures of 2.5 and 10 psi. (**d**) Water contact angles of SiO_2_/PVDF and boehmite/SiO_2_/PVDF membranes.

**Figure 6 polymers-15-02270-f006:**
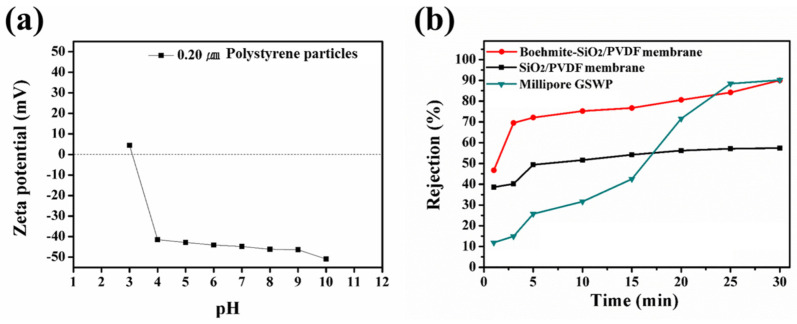
(**a**) Zeta potential of 0.20 μm polystyrene particles used for filtration tests at pH 3–10. (**b**) Rejection rates of polystyrene particles in SiO_2_/PVDF, boehmite-SiO_2_/PVDF composite, and Millipore GSWP membranes.

**Table 1 polymers-15-02270-t001:** Atomic ratio of the electrospun SiO_2_/PVDF membrane obtained from XPS.

	Binding Energy (eV)	Atomic Ratio (%)
Si 2p	103.7	24.44
C 1s	286.4	24.73
O 1s	533.2	44.41
F 1s	688.2	6.41

## Data Availability

Not applicable.

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
