# Peer review of "Electropositive Membrane Prepared via a Simple Dipping Process: Exploiting Electrostatic Attraction Using Electrospun SiO2/PVDF Membranes with Electronegative SiO2 Shell"

_polymers, 2023, doi:10.3390/polym15102270_

Round 1

Reviewer 1 Report

Manuscript: polymers-2371903

The manuscript describes the methodology to fabricate electropositive membrane based on Boehmite/SiO2 /PVDF in a straightforward manner. The manuscripts seems to fit within the scope of Polymers but before the reconsidering of the manuscript for publishing, it should be improved. Please refer to the comments below.

GENERAL COMMENTS

-the manuscript should be language-edited for the English language, style and punctuation

-the link between the scope of the Polymers journal should be more accented

ABSTRACT

-this part should be rewritten; at the moment it seems to be written more in the style intended for introduction rather than abstract; the Reviewer recommends to include one introductory sentence, identify the goal or niche that is addressed by the manuscript and to abstract the research presented

-“via” instead of “vis”

-please remove “rationally”

KEYWORDS

-the keywords should be edited to fit more the content of the manuscript, e.g. instead of “membrane”- “electropositive membrane”; the Reviewer suggests to include the intended application of the membrane in the keywords as well

INTRODUCTION

-this part should be rewritten to introduce content of the manuscript better; please consider including short additional fragments:

×          “membrane filtration following different mechanism other than physical sieving has recently gained lots of traction(…)” – please add examples of the other mechanisms and highlight the advantages of “electroactive filtration

×          “. Aluminium oxide is most widely utilized electropositive component for electropositive membrane fabrication because of its low toxicity and low cost (…)” please give examples of some other materials used (cf. previous comment)

×          please describe why alternative methods of boehmite deposition cannot be used in a straight forward manner (to strengthen the method the Authors used)

×          the Authors should mention the

-there are numerous statements that should be supported by a reference from the literature

-the fragment where the filtering out of negative species is not clear, please consider rephrasing;

-please remove “successfully” (at the end of the introduction)

- at the end of this section, the research reported in the manuscript should be introduced, i.e. the goal of the research, brief description what was investigated (to guide the Reader through the content) and a potential application of the material developed, should be stated (in the current form of the manuscript, this part is written in the conclusion style)

MATERIALS AND METHODS

-please add information about the purity and concentration of the chemicals used

-were the solutions diluted for the DLS measurements?

-XPS technical details and measurement protocols missing: what was the radiation source, the overall resolution of the spectromemeter, pass energy used for survey and region measuremens? was accumulation of the charge over the surface observed? if yes, what was the protocol for charge neutralization/correction? what was the protocol for the fitting of the spectra and the background? what software was used for the fitting of the spectra?

-how were the

RESULTS AND DISCUSSION

-all figure captions should be improved to introduce the figures in a more explicit manner; and example from the literature showing a detailed description both of the figures and an experimental section can be found here: Journal of Electroanalytical Chemistry, 1 (2019), p. 311

-there are some technical details (which should be moved to the experimental part) present in several please

-the Si2p spectrum seems to be asymmetric, pointing at the presence of other oxidation statef of Si

-“wide scan of XPS spectra” – please consider changing to “XPS survey spectr”a

-“ Though instant exposure is not critical to human beings, aluminum might cause a health issue when one is exposed to it for a long period of time.” – please add a reference

-p. 7 “As expected,(…)” please state why this result  was expected

-the manuscript should be language-edited for the English language, style and punctuation

-some longer phrases should be shortened to simpler grammatical construction, which would give the results more relevance and the description of the figures and graphs should be developed more in order to guide the Reader through the manuscript to the conclusion part.

Author Response

Responses to Reviewer #1

We thank the Reviewer for the comments and suggestions. Summarized below are our responses to the Reviewer’s specific concerns.

Comment 1:   The manuscript should be language-edited for the English language, style and punctuation

Response 1:   The revised manuscript was undertaken proofreading for English grammar, spell and correction by authors and professional English editing service (Enago). Please refer to the certificate below, thank you.

Comment 2:      The link between the scope of the Polymers journal should be more accented

Response 2:   The Abstract was more accented with the point of functional hybrid composites and surface modification in the revised manuscript.

Comment 3:   This part should be rewritten; at the moment it seems to be written more in the style intended for introduction rather than abstract; the Reviewer recommends to include one introductory sentence, identify the goal or niche that is addressed by the manuscript and to abstract the research presented.

Response 3:   Reviewer’s comment is appropriate and appreciated. Abstract was revised according to the reviewer’s comment as follow.

~ Most electropositive membranes are fabricated by decorating host membrane with electropositive particles. However, it requires complicated processes to securely attach electropositive particles on the host membrane surface. This study presents a simple dipping process for fabricating boehmite/SiO2/PVDF electropositive membranes by modifying the electrospun SiO2/PVDF host membrane using electropositive boehmite nanoparticles (NPs). of attaching boehmite on host membrane. By fabricating electrospun SiO2/PVDF host membrane with SiO2 shell having highly electronegative charge, we electrostatic attraction between the host membrane and the boehmite nanoparticle could be exploited to deposit electropositive boehmite nanoparticles by simply dipping in boehmite solution. The surface modification enhanced the filtration performance of the fabricated Boehmite/SiO2/PVDF electropositive of the membrane, as revealed by using electronegatively charged polystyrene (PS) NPs as a bacteria model. with 0.2 mm diameter For filtration performance comparison, commercial Millipore GSWP filter, with pore size of 0.2 mm, was rationally selected. The boehmite/SiO2/PVDF electropositive membrane, with an average pore size of 0.3 μm, could successfully filter out 0.2-μm PS particles. The rejection rate was comparable to that of Millipore GSWP, a commercial filter with a pore size of 0.2 μm, which can filter out 0.2 μm particles via physical sieving. In addition, the water flux of the boehmite/SiO2/PVDF electropositive membrane was twice that of Millipore GSWP, demonstrating the potential of the electropositive membrane in water purification and disinfection.

Comment 4:   “via” instead of “vis”

Response 4:   It was corrected in the revised manuscript.

Comment 5:   please remove “rationally”

Response 5:   It was removed in the revised manuscript.

Comment 6:   The keywords should be edited to fit more the content of the manuscript, e.g. instead of “membrane”- “electropositive membrane”; the Reviewer suggests to include the intended application of the membrane in the keywords as well.

Response 6:   Keywords are replaced as Electrospinning; electrostatic attraction; dipping; electropositive membrane; filter.

Comment 7:   This part should be rewritten to introduce content of the manuscript better; please consider including short additional fragments:

              “membrane filtration following different mechanism other than physical sieving has recently gained lots of traction(…)” – please add examples of the other mechanisms and highlight the advantages of “electroactive filtration

              “Aluminum oxide is most widely utilized electropositive component for electropositive membrane fabrication because of its low toxicity and low cost (…)” please give examples of some other materials used (cf. previous comment)

              please describe why alternative methods of boehmite deposition cannot be used in a straight forward manner (to strengthen the method the Authors used)

Response 7:    According to Reviewer’s comment, Introduction was written with following paragraph in the revised manuscript.

Currently, various techniques, such as mechanical and chemical methods, are being investigated for filtration. Water purification using bulk mechanical filters is the most common technique in water treatment. Sand, hydroanthracite, burned rocks, and crushed expanded clay are also used for filtration [7]. Additionally, the various types of adsorption membranes have been used to remove heavy metals and organic dyes from wastewater. Among them, metal-organic structures (MOFs) have been used in many fields owing to their high surface area and tunable pore volume and chemical properties, and MOF-mixed membranes containing nanoparticles (NPs) have been studied [8].

Polyelectrolytes [15], zirconia [16], copper oxide [17] and hematite [18] have been recently investigated as electropositive coatings. However, such electropositive coatings usually suffer from poor adhesion to the membrane, low zeta potential, and toxicity.

To date, boehmite deposition during the fabrication of electropositive membranes is achieved by direct hydrothermal synthesis on the host membrane [9]. Therefore, polymeric hosts cannot be used as they cannot withstand harsh hydrothermal conditions. A facile dipping process for boehmite deposition would enable the usage of various host membranes. In addition, the dipping process allows continuous fabrication, making it more cost effective than batch-type hydrothermal processes.

Comment 8:   There are numerous statements that should be supported by a reference from the literature

Response 8:   The proper references were updated in the revised manuscript.

Comment 9:  The fragment where the filtering out of negative species is not clear, please consider rephrasing;

Response 9:   The sentence was corrected in the revised manuscript as follow.

Therefore, ultrafine electronegative particles can be filtered through pores larger than the particles without sacrificing the flux of the feed.

Comment 10:   Please remove “successfully” (at the end of the introduction)

Response 10:  It was removed in the revised manuscript.

Comment 11: At the end of this section, the research reported in the manuscript should be introduced, i.e. the goal of the research, brief description what was investigated (to guide the Reader through the content) and a potential application of the material developed, should be stated (in the current form of the manuscript, this part is written in the conclusion style)

Response 11: Reviewer’s comment is appropriate and appreciated. Abstract was revised according to the reviewer’s comment as follow.

                          This research aims to develop a facile and cost-effective method for fabricating electropositive membranes for highly efficient water filtration.

~, demonstrating the potential of the electropositive membrane in water purification and disinfection.

Comment 12:   Please add information about the purity and concentration of the chemicals used.

Response 12:  In p.3, the purity information was added.

Comment 13:   Were the solutions diluted for the DLS measurements?

Response 13:  All the sample for the DLS measurements were diluted. Synthesized boehmite and 0.20 µm Polystyrene (PS) particles were dispersed in ethyl alcohol to make 100 ppm dispersion for DLS measurements. We added the details in the characterizations part as follow.

Dynamic laser scattering (DLS, Malvern Zetasizer) analysis was used to determine the zeta potential of the samples at different pH values and at the isoelectric point (IEP). The synthesized boehmite and 0.20-µm PS particles were dispersed in ethyl alcohol to make 100-ppm dispersion for the DLS measurements.

Comment 14: XPS technical details and measurement protocols missing: what was the radiation source, the overall resolution of the spectromemeter, pass energy used for survey and region measuremens? was accumulation of the charge over the surface observed? if yes, what was the protocol for charge neutralization/correction? what was the protocol for the fitting of the spectra and the background? what software was used for the fitting of the spectra?

Response 14:  As the reviewer suggested, details about the XPS measurement were added. Below is the revised version.

The atomic composition and presence of specific chemical bonds were studied using X-ray photoelectron spectroscopy (XPS) with an Al K-alpha X-ray source (Thermo Scientific K-alpha). For the survey scan, 200-eV pass energy was used, and the scanning was performed twice. For fine scans, 50-eV pass energy was used, and five scans were per-formed. All scans were executed with a flood gun to minimize the charge accumulation. The resultant spectra were analyzed using Advantage software provided with the XPS instrument.

Comment 15: All figure captions should be improved to introduce the figures in a more explicit manner; and example from the literature showing a detailed description both of the figures and an experimental section can be found here: Journal of Electroanalytical Chemistry, 1 (2019), p. 311

Response 15:  The captions of Figures were revised for more detailed description.

Comment 16: There are some technical details (which should be moved to the experimental part) present in several please.

Response 16:  The technical details which are not appropriate in Chap. 3.1 and 3.2 were removed in the revised manuscript.

Comment 17: The Si2p spectrum seems to be asymmetric, pointing at the presence of other oxidation state of Si.

Response 17:  As the reviewer mentioned, slight asymmetricity is observed but the degree of asymmetry is not significant, as the peak is mostly consisted of signals from SiO2. Further, as we simply used XPS data to obtain atomic weight percent ratio rather than deconvoluting XPS spectra to thoroughly distinguish different bonding states, we did not add detailed comments about the spectra shapes.

Comment 18:   “wide scan of XPS spectra” – please consider changing to “XPS survey spectra.

Response 18:  It was replaced according to Reviewer’s comment in the revised manuscript.

Comment 19: “Though instant exposure is not critical to human beings, aluminum might cause a health issue when one is exposed to it for a long period of time.” – please add a reference.

Response 19:  Ref [35] was added in the revised manuscript.

Comment 20:   p.7 “As expected,(…)” please state why this result was expected.

Response 20:  The sentence was modified for better readability in the revised manuscript as follow.

Due to the aforementioned hydrophilicity, the ~

Comment 21: The manuscript should be language-edited for the English language, style and punctuation.

Response 21:  The revised manuscript was undertaken proofreading for English grammar, spell and correction by authors and professional English editing service (Enago). Please refer to Response 1.

Comment 22: Some longer phrases should be shortened to simpler grammatical construction, which would give the results more relevance and the description of the figures and graphs should be developed more in order to guide the Reader through the manuscript to the conclusion part.

Response 22:  The manuscript was revised during proofreading for better readability.

Reviewer 2 Report

(1) In the manuscript entitled "Electropositive Membrane Prepared via Simple Dipping Process: Exploiting Electrostatic Attraction by Electrospun SiO2/PVDF Membrane with Electronegative SiO2 Shell ", the author proposes to apply electrostatic attraction to deposit electropositive boehmite nanoparticles by simply dipping in boehmite solution. The concept is good, but I think the innovations of this manuscript are still unclear for the following reasons:

1. The author should state what kind of scientific issues they hope to solve. Such as the key limitations for synthesizing the membrane assembly of electropositive filter. Following by providing their unique solution strategies to declare their innovations.

2. The authors hoped to filter the negatively charged particles such as bacteria and virus, why SiO2 is necessary for the host membrane. Why not directly use boehmite/PVDF components?

3. The performance of a material is highly controlled by its components and structure. What is the structure and composition of Millipore GSWP. Is it comparable with the Boehmite/SiO2/PVDF membrane?

3. ‘Boehmite/SiO2/PVDF electropositive membrane, whose average pore size was 0.3 μm, exhibited similar rejection rate when compared to the GSWP filter with 0.2 μm pore size the water flux was almost 2 times higher. Therefore, it could be concluded that Boehmite/SiO2/PVDF electropositive membrane exhibited sound membrane performance.’ I think the result is confusing. First, electropositive membrane with an average pore size of 0.3 um compared to the GSWP filter with an average pore size of 0.2, Cross-sectional area is 2.25 times higher. It is not a surprising result that the water flux is almost 2 times higher. In addition, the water flux of the prepared membrane performed an obvious decrease in 30 mins, while the commercial one still kept a stable performance. Moreover, no significant rejection rate enhancement was observed. It is difficult to convince others that the prepared membrane is better than the commercial one.

Therefore, the author should highlight his own original innovation from both scientific and practical concerns, including what are the characteristics of your material, why your method is better than others, etc.

(2) Kindly revised the current abstract. No need to use that large pages to introduce the synthesis process. The keywords are too simple, which could not represent the innovation and characteristic of your work.

(3) Kindly revised logical structure of introduction. No need to use too large pages for introducing the items not quite relevant to this study, such as physical sieving membrane filtration.

(4) Provide more scientific data. Such as the properties of boehmite solution? What is the concentration of boehmite in the solution. How the boehmite contents on membrane surface affect the membrane performance.

Minor comments

(1) Size/type of Fonts in each figure should be standardized and uniformed., size and the specifications of almost every pictur are different.

(2) No obvious difference present in Fig 5 a and b, Fig 2 a and b.

(3) Better to revise the current conclusion because it is too long to read and without the perspective of the material.

The English presentation is OK, but need to concern on the logical issue.

Author Response

Responses to Reviewer #2

We thank the Reviewer for the comments and suggestions. Summarized below are our responses to the Reviewer’s specific concerns.

Comment 1:   The author should state what kind of scientific issues they hope to solve. Such as the key limitations for synthesizing the membrane assembly of electropositive filter. Following by providing their unique solution strategies to declare their innovations.

Response 1:   The impact of present study was addressed in compare with previous works in the Conclusions of revised manuscript.

In existing surface-treatment methods, pores on the treated surfaces are blocked, thereby lowering the filtration efficiency. The proposed technique overcomes such a problem. The low water contact angle of the boehmite/SiO2/PVDF increases the flow rate and reduces energy consumption.

Comment 2:   The authors hoped to filter the negatively charged particles such as bacteria and virus, why SiO2is necessary for the host membrane. Why not directly use boehmite/PVDF components?

Response 2:   PVDF is also known as electronegative material, the zeta potential of PVDF is 0 ~ -30 mV (Polymers 2020, 12, 1303), while that of SiO2 is 0 ~ -120 mV (Scientific Reports 2016, 6, 22029). In this regard, SiO2 has strong negative-charge but still require PVDF the flexible material to being flexible as a host fiber.

Comment 3:   The performance of a material is highly controlled by its components and structure. What is the structure and composition of Millipore GSWP. Is it comparable with the Boehmite/SiO2/PVDF membrane?

‘Boehmite/SiO2/PVDF electropositive membrane, whose average pore size was 0.3 μm, exhibited similar rejection rate when compared to the GSWP filter with 0.2 μm pore size the water flux was almost 2 times higher. Therefore, it could be concluded that Boehmite/SiO2/PVDF electropositive membrane exhibited sound membrane performance.’

I think the result is confusing. First, electropositive membrane with an average pore size of 0.3 um compared to the GSWP filter with an average pore size of 0.2, Cross-sectional area is 2.25 times higher. It is not a surprising result that the water flux is almost 2 times higher. In addition, the water flux of the prepared membrane performed an obvious decrease in 30 mins, while the commercial one still kept a stable performance. Moreover, no significant rejection rate enhancement was observed. It is difficult to convince others that the prepared membrane is better than the commercial one. Therefore, the author should highlight his own original innovation from both scientific and practical concerns, including what are the characteristics of your material, why your method is better than others, etc.

Response 3:   We really appreciate your suggestion. Originally, we intended to highlight the rejection of 0.20 mm particles via boehmite/SiO2/PVDF electropositive membranes with 0.30 mm pore size. Conventional membranes following physical sieving mechanism, including a GSWP filter, cannot filter out the particles smaller than their pore sizes. However, electropositive membranes can filter out smaller particles than their pore sizes, therefore they exhibit higher water flux than conventional counterparts. As suggested, we revised the part where the reviewer pointed out, as follows.

‘Boehmite/SiO2/PVDF electropositive membrane, whose average pore size was 0.3 μm, exhibited similar rejection rate when compared to the GSWP filter with 0.2 μm pore size the water flux was almost 2 times higher. Therefore, it could be concluded that Boehmite/SiO2/PVDF electropositive membrane exhibited sound membrane performance.’

The boehmite/SiO2/PVDF electropositive membrane, with an average pore size of 0.30 μm, could successfully filter out 0.20-μm PS particles. The rejection rate was comparable to that of Millipore GSWP, a commercial filter with a pore size of 0.22 μm, which can filter out 0.20 μm particles via physical sieving. In addition, the water flux of the boehmite/SiO2/PVDF electropositive membrane was twice that of Millipore GSWP,

Comment 4:   Kindly revised the current abstract. No need to use that large pages to introduce the synthesis process. The keywords are too simple, which could not represent the innovation and characteristic of your work.

Response 4:   In the Introduction, abundant sentences were removed in the revised manuscript as follow. Keywords are replaced as Electrospinning; electrostatic attraction; dipping; electropositive membrane; filter.

                        By fabricating electrospun SiO2/PVDF host membrane with SiO2 shell having highly electronegative charge [3,4], we could easily attach electropositive boehmite through short time dipping process of host membrane in boehmite solution through electrostatic attraction. Attachment of nanoparticles via electrostatic interaction was demonstrated in previous articles [5-7] and boehmite was successfully attached to the host membrane.

Comment 5:   Kindly revised logical structure of introduction. No need to use too large pages for introducing the items not quite relevant to this study, such as physical sieving membrane filtration.

Response 5:   Introduction was strengthened include relevant literature review of SiO2/PVDF nanofiber in the revised manuscript as follow.

PVDF-g-PNE and PVDF-g-PAA membranes contain electropositive materials; thus, bacteria and viruses, which normally exhibit negative charges, can be effectively filtered via electrostatic attraction. Polymer coatings firmly bond to PVDF membranes through adhesive force (coordination, hydrogen bonding, electrostatic interaction, and hydro-phobic interaction) [25]. SiO2 NPs have high electronegativity (2.82) and attract electro-positive lithium ions from electrolytes. On this basis, a SiO2/PVDF composite membrane has been developed as a battery separator by varying the SiO2-to-PVDF mass ratio [26]. Graft copolymerization of methacrylic acid (MAA) monomers with plasma-activated PVDF membranes was performed to introduce carboxyl groups into the membrane. Subsequently, the surface of the NPs was made hydrophilic using a positively charged ligand, and the NPs were coated on the membrane surface through electrostatic and covalent bonding [27].

Comment 6:   Provide more scientific data. Such as the properties of boehmite solution? What is the concentration of boehmite in the solution. How the boehmite contents on membrane surface affect the membrane performance.

Response 6:   11.5 wt% of boehmite solution was used as dipping solution in this study. Definitely, the higher the boehmite content, the better the thermal characteristics and the higher the adsorption performance of heavy metal ions. Accordingly, it can be seen that a higher boehmite content gives a higher surface potential and brings more charge [8,9].

Comment 7:   Size/type of Fonts in each figure should be standardized and uniformed., size and the specifications of almost every picture are different.

Response 7:   The fonts in each Figures are corrected for better readability in the revised manuscript.

Comment 8:   No obvious difference present in Fig 5 a and b, Fig 2 a and b.

Response 8:   Reviewer’s comment is appropriate and appreciated, thus Figure 2a and b was removed in the revised manuscript.

Comment 9:   Better to revise the current conclusion because it is too long to read and without the perspective of the material.

Response 9:   The abundant contents were removed and impact of present study was addressed in the Conclusions of revised manuscript as follow.

Strategy was to first fabricate electrospun membrane with electronegative shell so that electrostatic attraction between electropositive boehmite and electronegative surface can assist boehmite attachment to the surface. So, mixture of electronegative SiO2 and polymer PVDF was rationally selected to be electrospun into non-woven membrane to take advantage of phase separation between SiO2 and PVDF in a way that SiO2 migrates toward skin part. As a result, multi core nanofiber having well defined SiO2 shell could be electrospun and collected to form non-woven membrane. SiO2/PVDF membrane was then simply submerged in boehmite solution for boehmite attachment. Surprisingly, boehmite successfully covered entire surface of SiO2 /PVDF through simple short time dipping process.

In existing surface-treatment methods, pores on the treated surfaces are blocked, thereby lowering the filtration efficiency. The proposed technique overcomes such a problem. The low water contact angle of the boehmite/SiO2/PVDF increases the flow rate and reduces energy consumption.

Comment 10: The English presentation is OK, but need to concern on the logical issue.

Response 10:  We have rearranged the logical structure by focusing on the key strategy and objective of this study Including answers to comments 2 and 3. In particular, unnecessary sentences were removed from the introduction and conclusion, and related content was added to smooth out the flow.

Reviewer 3 Report

Publish after major revisions noted below:

1.      Introduction should be separated into at least three paragraphs.

2.      Attachment of nanoparticles via electrostatic interaction on SiO2/PVDF nanofiber membrane should be thoroughly reviewed in the introduction.

3.      Figure 5, the SEM images of (a) electrospun SiO2/PVDF membrane and (b) Boehmite-SiO2/PVDF composite membrane fabricated by dipping are the same. The author might have put the wrong SEM for Boehmite-SiO2/PVDF.

4.      The authors should point out the impact of their studies against other known arts in the literature.

5.      It is also important to prove if the Boehmite-SiO2/PVDF is reusable, and if the separation efficiency is stable.

Author Response

Responses to Reviewer #3

We thank the Reviewer for the comments and suggestions. Summarized below are our responses to the Reviewer’s specific concerns.

Comment 1:   Introduction should be separated into at least three paragraphs.

Response 1:     Introduction was separated into three paragraphs in the revised manuscript.

Comment 2:   Attachment of nanoparticles via electrostatic interaction on SiO2/PVDF nanofiber membrane should be thoroughly reviewed in the introduction.

Response 2:   The studies on SiO2/PVDF nanofiber were reviewed in the revised manuscript p.2 as follow.

PVDF-g-PNE and PVDF-g-PAA membranes contain electropositive materials; thus, bacteria and viruses, which normally exhibit negative charges, can be effectively filtered via electrostatic attraction. Polymer coatings firmly bond to PVDF membranes through adhesive force (coordination, hydrogen bonding, electrostatic interaction, and hydro-phobic interaction) [25]. SiO2 NPs have high electronegativity (2.82) and attract electro-positive lithium ions from electrolytes. On this basis, a SiO2/PVDF composite membrane has been developed as a battery separator by varying the SiO2-to-PVDF mass ratio [26]. Graft copolymerization of methacrylic acid (MAA) monomers with plasma-activated PVDF membranes was performed to introduce carboxyl groups into the membrane. Subsequently, the surface of the NPs was made hydrophilic using a positively charged ligand, and the NPs were coated on the membrane surface through electrostatic and covalent bonding [27].

Comment 3:   Figure 5, the SEM images of (a) electrospun SiO2/PVDF membrane and (b) Boehmite-SiO2/PVDF composite membrane fabricated by dipping are the same. The author might have put the wrong SEM for Boehmite-SiO2/PVDF.

Response 3:   It was carefully confirmed, and then explained once again as follow in p,6.

The SiO2/PVDF fibers showed a smooth surface (panel a), whereas, an irregular texture was observed on the fiber surface (panel b), confirming that boehmite coated the SiO2/PVDF fibers.

Comment 4:   The authors should point out the impact of their studies against other known arts in the literature.

Response 4:  The impact of present study was addressed in compare with previous works in the Conclusions of revised manuscript.

In existing surface-treatment methods, pores on the treated surfaces are blocked, thereby lowering the filtration efficiency. The proposed technique overcomes such a problem. The low water contact angle of the boehmite/SiO2/PVDF increases the flow rate and reduces energy consumption.

Comment 5:   It is also important to prove if the Boehmite-SiO2/PVDF is reusable, and if the separation efficiency is stable.

Response 5: We do agree that the stability of the membrane should be checked. Thanks for the suggestion. To prove the stability of the membrane, in Figure 5-c we flowed 100ml, 300ml, and 500ml of deionized water, respectively, and analyzed the filtrate with ICP-OES. With ICP-OES, detachment of Boehmite, which is contains aluminum atom, can be detected. As a result, no aluminum atom was detected upon operation (under 0.1 ppm, which is a detection limit of the ICP), thereby proving stability of the stability of the electropositive membrane fabricated by dip-coating process. In the revised manuscript, the following are added.

To check the amount of boehmite detached from the boehmite/SiO2/PVDF composite membrane during filtration, the filtrate was analyzed using inductively coupled plasma-optical emission spectroscopy (ICP-OES, iCAP 6000). Furthermore, 100, 300, and 500 ml of deionized water were filtered through the boehmite/SiO2/PVDF composite mem-brane, and the aluminum contents of filtrates were measured.

Round 2

Reviewer 2 Report

I think the quality of manuscript is well improved, which could be considered for publication.

(1) Kindly remove the marks that stated the revisions in the whole manucript, such as (R1-11) in line 15.

(2)  Line 49-line 57 needs more references to evidancing. Better to cite https://doi.org/10.1016/j.jclepro.2023.136612     in line 54

(3) No need to repeat the whole experiment design in conclusion. Just state the key issues this paper going to solve, representative results and perspectives of this research. 

(4) Carefully go through the whole mansucript and check if there is any mistakes, for example,  figure 3 was not mentioned in test.

Need to check, and be improved.

Author Response

Responses to Reviewer #2

We thank the Reviewer for the comments and suggestions. Summarized below are our responses to the Reviewer’s specific concerns.

Comment 1:   Kindly remove the marks that stated the revisions in the whole manuscript, such as (R1-11) in line 15.

Response 1:   The revision marks are removed in the revised manuscript.

Comment 2:   Line49-line57 needs more references to evidencing. Better to cite https://doi.org/10.1016/j.jclepro.2023.136612 in line 54.

Response 2:   Ref [8] is added in the revised manuscript.

Comment 3:   No need to repeat the whole experiment design in conclusion. Just state the key issues this paper going to solve, representative results and perspectives of this research.

Response 3:   The abundant sentences are removed in the revised manuscript.

Finally, particle rejection tests were performed using 0.20-mm PS NPs with highly negative charges to simulate bacteria filtration.

Comment 4:   Carefully go through the whole manuscript and check if there is any mistakes, for example, figure 3 was not mentioned in test.

Response 4:   It was carefully confirmed, and the typo (Figure 3) and corrected in the revised manuscript.

Reviewer 3 Report

Again, the SEM images in Figure 4 shows that a and b have similar smooth surface instead of an irregular texture for Figure 4b. Please double check. Besides this, the author revises the paper carefully based on the comments. The paper is ready to publish.

Author Response

Responses to Reviewer #3

We thank the Reviewer for the comments and suggestions. Summarized below are our responses to the Reviewer’s specific concerns.

Comment 1:   Again, the SEM images in Figure 4 shows that a and b have similar smooth surface instead of an irregular texture for Figure 4b. Please double check.

Response 1:   As seen in the Figure 4, the SiO2/PVDF fiber (panel a) shows a smooth and clean surface, whereas, an irregular texture with wrinkles is observed due to the boehmite-coat on it (panel b). This is clear evidence that boehmite is deposited and covered the host fiber in visual.
